Ecological and Evolutionary Science

# Fungal Communities Are Important Determinants of Bacterial Community Composition in Deadwood

Iñaki Odriozola,[a] Nerea Abrego,[b] Vojtěch Tláskal,[a] Petra Zrůstová,[a] Daniel Morais,[a] Tomáš Větrovský,[a] Otso Ovaskainen,[c] Petr Baldrian[a]

[a]Laboratory of Environmental Microbiology, Institute of Microbiology of the Czech Academy of Sciences, Prague, Czech Republic
[b]Department of Agricultural Sciences, University of Helsinki, Helsinki, Finland
[c]Organismal and Evolutionary Biology Research Programme, University of Helsinki, Helsinki, Finland

**ABSTRACT** Fungal-bacterial interactions play a key role in the functioning of many ecosystems. Thus, understanding their interactive dynamics is of central importance for gaining predictive knowledge on ecosystem functioning. However, it is challenging to disentangle the mechanisms behind species associations from observed co-occurrence patterns, and little is known about the directionality of such interactions. Here, we applied joint species distribution modeling to high-throughput sequencing data on co-occurring fungal and bacterial communities in deadwood to ask whether fungal and bacterial co-occurrences result from shared habitat use (i.e., deadwood's properties) or whether there are fungal-bacterial interactive associations after habitat characteristics are taken into account. Moreover, we tested the hypothesis that the interactions are mainly modulated through fungal communities influencing bacterial communities. For that, we quantified how much the predictive power of the joint species distribution models for bacterial and fungal community improved when accounting for the other community. Our results show that fungi and bacteria form tight association networks (i.e., some species pairs co-occur more frequently and other species pairs co-occur less frequently than expected by chance) in deadwood that include common (or opposite) responses to the environment as well as (potentially) biotic interactions. Additionally, we show that information about the fungal occurrences and abundances increased the power to predict the bacterial abundances substantially, whereas information about the bacterial occurrences and abundances increased the power to predict the fungal abundances much less. Our results suggest that fungal communities may mainly affect bacteria in deadwood.

**IMPORTANCE** Understanding the interactive dynamics between fungal and bacterial communities is important to gain predictive knowledge on ecosystem functioning. However, little is known about the mechanisms behind fungal-bacterial associations and the directionality of species interactions. Applying joint species distribution modeling to high-throughput sequencing data on co-occurring fungal-bacterial communities in deadwood, we found evidence that nonrandom fungal-bacterial associations derive from shared habitat use as well as (potentially) biotic interactions. Importantly, the combination of cross-validations and conditional cross-validations helped us to answer the question about the directionality of the biotic interactions, providing evidence that suggests that fungal communities may mainly affect bacteria in deadwood. Our modeling approach may help gain insight into the directionality of interactions between different components of the microbiome in other environments.

**KEYWORDS** biotic interactions, co-occurrence, cross-validation, conditional cross-validation, fungal-bacterial interactions, HMSC, joint species distribution modeling

Address correspondence to Iñaki Odriozola, inaki.odriozola@biomed.cas.cz.

By applying JSDM to fungal and bacterial communities, we show that fungal communities may mainly affect bacteria in deadwood. Our approach may help exploring the directionality of interactions between the components of the microbiome in other environments.

Fungi and bacteria are core members of communities driving biogeochemical cycles, and interactions between the groups play a key role in the functioning of numerous ecosystems (1). Thus, understanding their interactive dynamics is of central importance for gaining predictive knowledge on ecosystem functioning. Biotic interactions are one of the main assembly processes and are expected to result in nonrandom co-occurrence patterns between species (2). Interactive relationships such as mutualism, parasitism, and facilitation are expected to lead to aggregated distributions between species and positive species-to-species associations, whereas competition can be expected to lead to segregated distributions and negative species-to-species associations (3). However, inferring biotic processes from co-occurrence patterns in observational studies is not trivial: nonrandom patterns may result from shared habitat use rather than from interactive effects (3), whereas opposite responses to simultaneous environmental constraints may result in random co-occurrence patterns (4). Although only a manipulative experiment could establish a causal relationship between observed patterns and underlying biotic processes, recent advances in joint species distribution modeling (JSDM) (5) have significantly advanced the research on biotic interactions in observational studies by decomposing species co-occurrence patterns into shared environmental responses and residual species associations (3, 6, 7).

Fungi are the main contributors to wood decomposition and deeply modify its physical structure (8); thus, deadwood decomposition in forest ecosystems has been traditionally attributed to wood-decaying fungi. Nevertheless, the role of bacteria, either directly or through interactions with fungi, is being increasingly recognized (8–10). There are innumerable mechanisms by which fungi can affect bacteria and bacteria can affect fungi during wood decomposition, and studying them separately completely neglects such interactive effects (11). Fungi can strongly modify the deadwood environment by modifying pH or translocating N and P from soil (12, 13), which, in turn, may impact bacterial community composition (14, 15). Similarly, bacterial activity may alter wood properties through, for example, the fixation of atmospheric $N_2$ (16). Moreover, fungal-bacterial interactions go beyond these effects through modulation of the environment. There is ample *in vitro* evidence of antagonistic effects in both directions (17–19). Bacteria are able to directly feed on fungal mycelia (20). Bacteria can act as commensalists by consuming fungal exudates and products of wood decomposition (21, 22). Mutualistic interactions have also been described, where fungi gain protection against fungicides and bacteria get increased access to resources (23).

The capacity of fungi to modify the spatial structure of deadwood makes them crucial drivers of microbial community composition and activity (24). Mycelia enable fungi to exploit and expand through the three-dimensional space of deadwood (1), making hyphae an effective dispersal vector also for the wood-inhabiting bacteria (25). In this line, in experimental setups, Johnston et al. (26) and Christofides et al. (27) demonstrated strong directional effects from fungi to bacteria: they inoculated wood pieces with known wood-decaying fungi and then observed that the development of succeeding bacterial communities highly depended on the identity of the inoculated fungi. Therefore, fungal-bacterial interactions might be mainly modulated through fungal communities influencing bacterial communities.

Recent studies have shown that fungi and bacteria co-occur nonrandomly in deadwood (10, 28–30). However, using raw co-occurrences, none of the studies could distinguish between associations derived from shared habitat use and interactive effects nor could they explore the directionality of the interactions. Therefore, it remains largely unknown which are the principal mechanisms behind the reciprocal effects between fungi and bacteria (11).

The first objective of the present paper was to assess whether fungal and bacterial co-occurrences result from shared habitat use (i.e., deadwood's properties) or whether there are fungal-bacterial interactive associations after habitat characteristics are taken into account. For this, we applied JSDM to high-throughput sequencing data on co-occurring fungi and bacteria. Fungal community composition has been reported to be

strongly influenced by host tree species and deadwood diameter and decay stage (31–33), variables which reflect different physical-chemical properties. Likewise, bacterial community composition in deadwood is influenced by variables such as water content, pH, or C/N ratio of deadwood (26, 34–36). Hence, a wide range of log physical and chemical characteristics were included as fixed effects in the models, and residual association networks were compared between the models that did versus those that did not account for the response of species to the environmental variables. Additionally, we hypothesized that the fungal-bacterial interactions are mainly modulated through fungal communities influencing bacterial communities. To test the effects of fungal communities on the bacterial communities and the effect of bacteria on fungi, we quantified how much the predictive power of the JSDM for bacterial community improved when accounting for fungal community data and, vice versa, how much predictive power increased for the fungal community when accounting for bacterial community composition.

## RESULTS

Nonmetric multidimensional scaling (NMDS) ordinations show that the filters to remove rare operational taxonomic units (OTUs) altered overall community structure very little, and, similarly, the set of wood physical and chemical characteristics that significantly correlated with main trends in community structure also remained the same (Fig. 1). Fungal and bacterial species richness were uncorrelated (see also Fig. S1 in the supplemental material), whereas beta-diversity metrics of both domains were significantly correlated, regardless of the filtering criteria used (see Fig. S2). Therefore, we proceeded to hierarchical modeling of species communities (HMSC) using the community data sets filtered to bigger data sets (452 fungal and 570 bacterial OTUs) and smaller data sets (103 fungal and 51 bacterial OTUs).

From the species-to-species association matrices, there was evidence for both co-occurrences arising from shared habitat preferences and from interactive effects. The association matrix derived from the residuals of the null model was strongly structured both within and between fungi and bacteria, with several species co-occurring more and less often than expected by chance (Fig. 2A; Fig. S3A). In the residual association matrix of the full model (i.e., the associations remaining after the effects of environmental predictors were taken into account), several of those associations remained, but many others disappeared. This means that many of the species-to-species associations detected by the null model were the result of many species pairs responding in the same (or opposite) way to the characteristics of deadwood (Fig. 2B; Fig. S3B). Additionally, many other associations shifted from positive to negative or from negative to positive (changes in the direction of the associations are easiest to see in Fig. S3, since the association matrix involves a lower number of OTUs).

Variance partitioning community composition revealed that for bacteria, log-level random effects (i.e., the species-to-species association network) explained a greater fraction of the variance than those for fungi (Fig. 3; Fig. S4). The partitioning further showed that the fractions explained by log chemical characteristics in comparison to log physical characteristics were similar in both fungi and bacteria: chemistry explained more than double the variance than physical characteristics for both domains (Fig. 3; Fig. S4).

The models had generally somewhat greater predictive power for bacteria than for fungi, especially when the prediction was conditioned by the occurrences and abundances of fungi (Table 1; Table S1). For prediction of fungal occurrences (the binomial model), including the fixed effects (all environmental predictors) increased the predictive power by 0.031 area under the curve (AUC) points in comparison to the null model not including environmental predictors. Including also the information on bacterial occurrences and abundances increased the power an additional 0.014 AUC points (conditional predictive power of full model versus nonconditional predictive power of full model) (Table 1; Table S1). Bacterial community was nearly half as good a predictor of fungal occurrences than were the environmental predictors (i.e., log chemical and

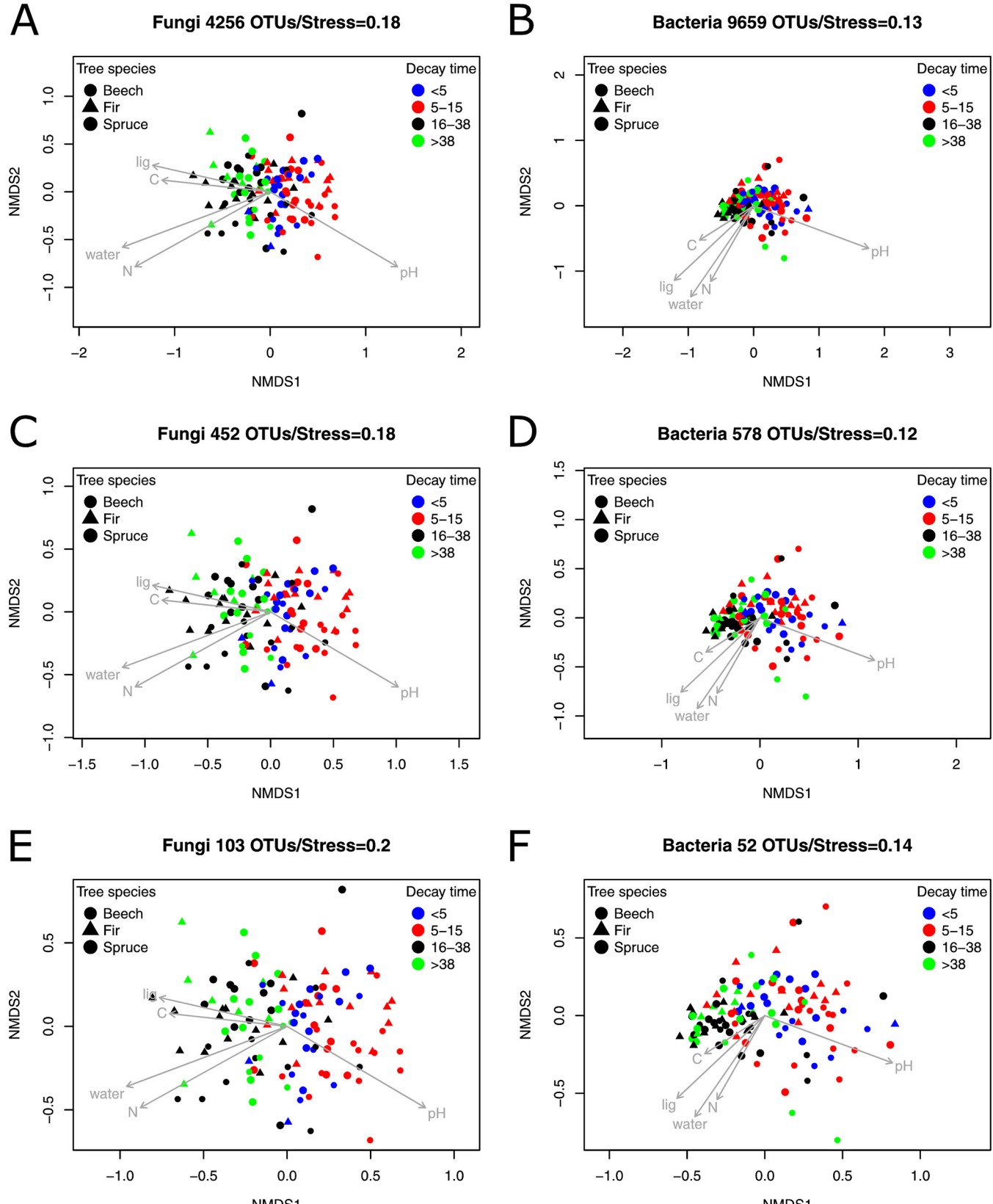

**FIG 1** Fungal and bacterial community structures with the whole community and after filtering rare OTUs. NMDS ordination plots showing the main trends in fungal community structure including all OTUs (A), after applying the filter resulting in bigger data set (C), and after applying the filter resulting in smaller data set (E). Bacterial community structure including all OTUs (B), after applying the filter resulting in bigger data set (D), and after applying the filter resulting in smaller data set (F). Gray arrows depict the environmental variables significantly associated with the main trends in community structure.

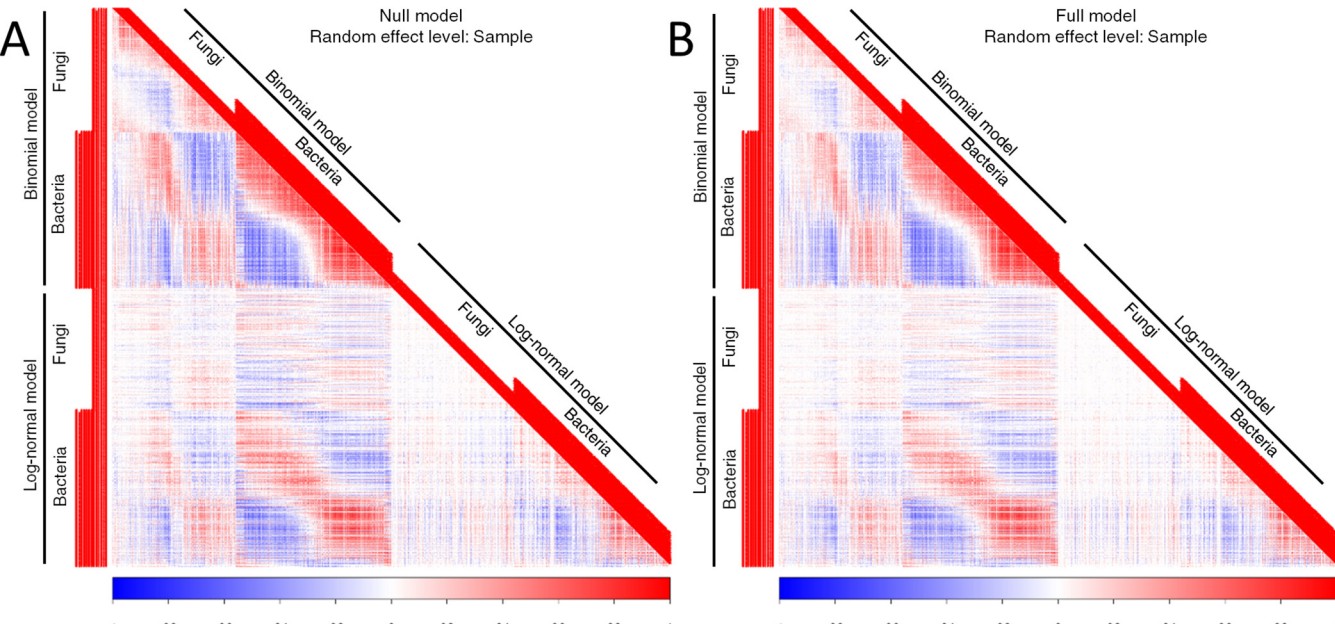

**FIG 2** Species association networks. Species association networks derived from the log (i.e., sample)-level random effects of the JSDM. (A) The network derived from the null model (i.e., including only sequencing depths as explanatory variables) shows raw species co-occurrences. (B) The network derived from the full model (i.e., including all measured environmental predictors) shows co-occurrences once shared (or opposite) habitat use has been taken into account. Differences between networks of null and full models are more visible in Fig. S3 in the supplemental material, since the association matrix involves lower number of OTUs.

physical characteristics). When predicting bacterial occurrences, environmental predictors increased predictive power by 0.031 AUC points, and including fungal occurrences and abundances increased the power an additional 0.059 AUC points (Table 1; Table S1). For bacterial occurrences, fungal community was nearly twice as good a predictor as were the environmental predictors. Prediction of abundances (the log-normal models) was overall worse, but the trends were similar: information about the fungal occurrences and abundances increased the power to predict the bacterial abundances substantially, whereas information about the bacterial occurrences and abundances increased the power to predict the fungal abundances much less (Table 1; Table S1).

## DISCUSSION

Fungal-bacterial interactions are ubiquitous in nature and play a key role in the function of many ecosystems (1). Here, we jointly studied fungal and bacterial communities in deadwood, and similar to other studies that reported nonrandom co-occurrence patterns between them (28–30), we observed that the species association network is strongly structured both within and between fungi and bacteria. Previous works have attributed the nonrandom associations between fungi and bacteria to modifications of wood chemistry by the fungal activity (26, 29); however, our results show that many significant associations remain after accounting for the effects of environmental predictors. This suggests that biotic interactions beyond the effects on and responses to environmental factors occur between fungi and bacteria.

In this study, we measured a wide range of environmental predictors, including both wood chemical and physical characteristics. Studies analyzing fungal communities in deadwood usually measure physical characteristics of wood, which, in turn, are correlated with chemistry (31–33), whereas wood chemical explanatory variables are more commonly included in studies involving the bacterial community (34–36). Here, addressing fungi and bacteria together, we show that both kinds of variables explained a similar fraction of variance in these domains: chemistry was twice as important as log physical characteristics. Since we lack temporal data, we could not distinguish whether fungi modified the wood chemistry first and bacteria responded to the modification or

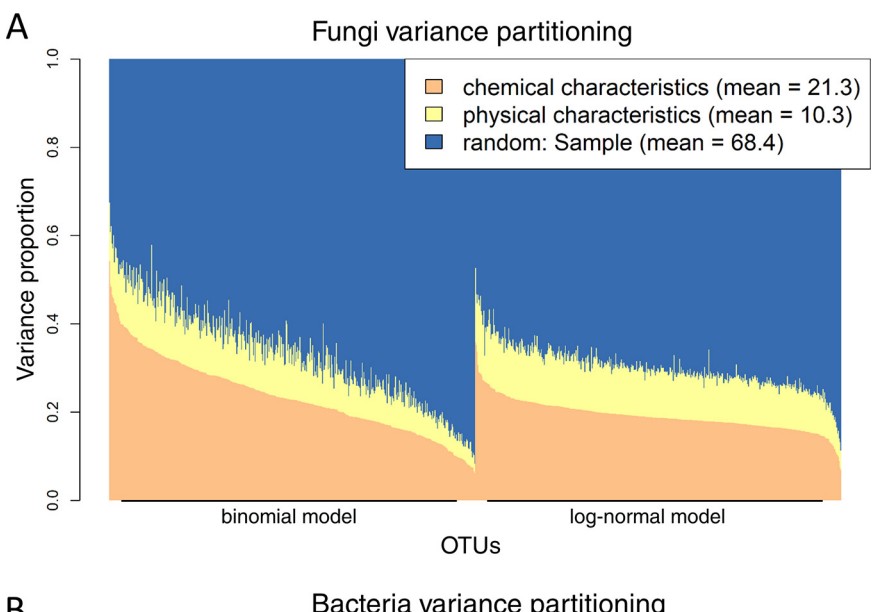

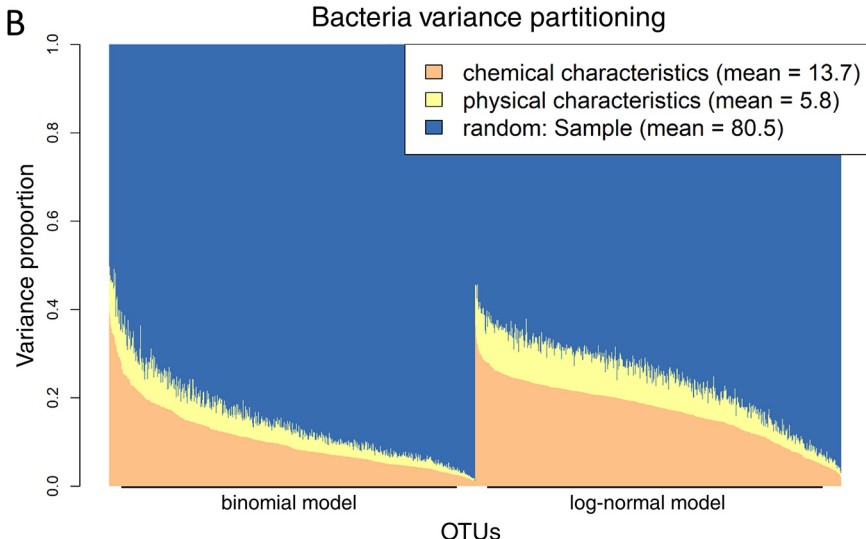

**FIG 3** Variance partitioning of fungal and bacterial community compositions. Proportions of variance explained by log chemical (water, pH, C, N, and lignin) and physical characteristics (tree species identity, decay time, and DBH) and sample-level random effects (i.e., species-to-species association matrix) in fungi (A) and bacteria (B). Bar plots show the variance proportions species by species, whereas text boxes show proportions averaged over species and models (binomial versus log normal). Species are differently ordered in binomial and log-normal parts of the model, since they were separately ordered based on the effect size of chemical predictors. Note that only explained variance is depicted in the plot: the explanatory powers of binomial and log-normal models for fungi and bacteria are reported in the corresponding column of Table 1.

if both bacteria and fungi are responding to wood chemistry, which, in turn, depends on other log characteristics. The answer is probably complex, since dominant fungi may shape subsequent bacterial communities by lowering wood pH (13, 26), but different tree species may have specific chemical characteristics as well (33, 34).

Variance partitioning revealed a stronger effect of log-level random effects on bacterial community composition than on fungal community composition. Log-level random effects likely model interactive associations between species, since environmental predictors were included in the JSDM as fixed effects; however, they could also represent missing environmental covariates. In the same line, cross-validation results suggest directional interactive effects from fungi to bacteria: information on fungal occurrences

**TABLE 1** Explanatory and predictive powers of the models[a]

| Category | Model | No. of species | Explanatory power | Unconditional predictive power | Conditional predictive power |
|---|---|---|---|---|---|
| Occurrence | | | | | |
| Fungi | Null | 452 | 0.786 | 0.568 | 0.607 |
| | Full | 452 | 0.824 | 0.599 | 0.613 |
| Bacteria | Null | 570 | 0.876 | 0.556 | 0.665 |
| | Full | 570 | 0.894 | 0.587 | 0.646 |
| Abundance | | | | | |
| Fungi | Null | 452 | 0.315 | −0.051 | −0.016 |
| | Full | 452 | 0.425 | −0.010 | 0.001 |
| Bacteria | Null | 570 | 0.398 | −0.002 | 0.049 |
| | Full | 570 | 0.486 | 0.012 | 0.047 |

[a]Explanatory and predictive powers of the models for fungal and bacterial occurrences (measured in terms of AUC of the binomial model) and abundances (measured in terms of $R^2$ of the log-normal model) were averaged over the species. Null models have sequencing depth as the sole explanatory variable, whereas full models include also the environmental predictors. Explanatory power is based on model fitted to all data, unconditional predictive power is based on 2-fold cross-validation, where values of environmental predictors are known, and conditional predictive power is based on 2-fold cross-validation where the occurrences and abundances of the nonfocal group (bacteria to predict fungi and fungi to predict bacteria) are also assumed to be known. Note that AUC index takes values between 0.5 and 1, whereas $R^2$ ranges between 0 and 1.

and abundances was twice as good a predictor of bacterial community composition than the environmental predictors altogether. However, this was not the case when using information on bacterial communities to predict fungi; in this case, environmental predictors had twice as good predictive power. Again, it might be argued that unmeasured explanatory variables rather than actual biotic interactions are responsible for residual associations between fungi and bacteria. We believe this to be unlikely, because a common response of fungi and bacteria to unmeasured variables would have resulted in symmetrical effects between fungi and bacteria, which is not the case in our study—fungi are a good predictor of bacteria, but the opposite is not true. The overall low predictive powers of the models are not surprising given the inherent stochasticity of microbial community development. There are many well-documented mechanisms affecting fungal and bacterial community assembly and altering their spatial distributions, such as drift, dispersal limitation, priority effects, or endpoint assembly cycles (37). Moreover, it is challenging predicting highly diverse communities with a relatively small data set.

The directional effect of fungi on bacteria is in line with other studies (26, 27, 29), which suggested that the modification of pH by fungi is an important underlying mechanism. Even if we cannot give any final answer on the involved mechanisms with this observational study, our results suggest that part of the effect of fungi on bacteria is uncorrelated with any chemical variable of the wood. There are alternative ways fungi might affect bacterial composition. Fungal hyphae are proficient in exploring the wood three-dimensional space, and they can be considered to constitute ideal transport paths and scaffolds for bacteria (1). In fact, bacteria have been reported to use fungal mycelia to disperse more efficiently in other substrates (23, 38).

The joint movement of bacteria with fungi (and hence, their interactions) might have deep consequences for deadwood decomposition and forest nutrient dynamics. Previous findings on co-occurrence patterns between wood-decaying fungi and $N_2$-fixing bacteria suggest that deadwood decomposition might be an interactive process where bacteria may provide the N source and fungi provide the C source (10, 11, 28, 36). Similar interactions between fungi and bacteria have also been reported in the process of litter decomposition (38). Furthermore, implications of fungal-bacterial associations in ecosystem functions go well beyond plant matter decomposition (1). For instance, mycelium-based dispersal improves the movement of bacteria in heterogeneously polluted soils where movement of bacteria is otherwise impaired, thus stimulating contaminant biodegradation (39–41).

mSystems®

Finally, the JSDM approach that we applied does not model explicitly compositional data using a multinomial distribution. Extending the data models of JSDMs to include the multinomial distribution is one of the key challenges in the ongoing merging of multivariate methods developed separately for microbial ecology and for the community ecology of macroorganisms (3). Failing to account for the multinomial nature of the data could theoretically generate negative associations between species. This would certainly be the case if the data involve two species, as then the increase in sequence count of one species would directly decrease the sequence count of the other species. However, even for the case of two species, the presence-absence part of the model would be less problematic, as missing the occurrence of one species because the other species dominates the sequence data would be unlikely. Moreover, we believe that this is not a major problem for the abundance data either, since the community consists of a very large number of species, none of which dominates the data in terms of sequence abundance. Thus, the constraining effect of the total number of sequences is diluted through the whole community and is not likely to generate strong negative associations between any pair of species. Reflecting this, the associations we report are not predominantly negative (Fig. 2), but they contain a balanced set of associations that are positive, negative, or not statistically supported. Therefore, we consider our results robust even if the JSDM applied here does not include the multinomial data model.

In conclusion, our study shows that fungi and bacteria form tight association networks (i.e., they co-occur more or less frequently than expected by chance) in deadwood that include common (or opposite) responses to the environment, as well as (potentially) biotic interactions. Importantly, the combination of cross-validations and conditional cross-validations helped us to answer the question about the directionality of the biotic interactions, providing observational evidence suggesting that fungal-bacterial interactions may be modulated through fungal communities influencing bacterial communities. Our modeling approach may help gaining insight into the directionality of interactions between different components of the microbiome in other environments.

## MATERIALS AND METHODS

**Study area and sampling design.** The study area was located in the 25-ha Zofin ForestGEO Dynamics plot in the Novohradské Hory mountains, Czech Republic (48°39′57′′N, 14°42′24′′E; www.forestgeo.si.edu). This area is part of the 42-ha core zone of the Žofínský prales National Nature Reserve (established 1838), which has never been managed and thus represents a virgin forest. The bedrock consists of fine- to medium-grain porphyritic and biotite granite. Annual average rainfall is 866 mm, and annual average temperature is 6.2°C (42). The living tree volume, which is calculated to be 690 m³ ha⁻¹ (43), is dominated by *Fagus sylvatica* (51.5% of total living wood volume), followed by *Picea abies* (42.8%) and *Abies alba* (4.8%). Other tree species (*Ulmus glabra*, *Acer pseudoplatanus*, *Acer platanoides*, and *Sorbus aucuparia*) represent 1% of living wood volume. The dead coarse wood debris is calculated to be on average 208 m³ ha⁻¹ (44) and is more evenly represented by *F. sylvatica*, *P. abies*, and *A. alba*, with 23.6%, 43.7%, and 31.4% of the volume, respectively (43).

The log sampling scheme was detailed in previous publications (33, 36), but here follows a brief description. For the study area, detailed information is available about every living and dead tree with diameter at breast height (DBH) of ≥10 cm (including spatial location, DBH, tree species, tree status, live/dead, standing/lying, snag, breakage, windthrow, stump, etc.): all the variables were repeatedly recorded in 1975, 1997, 2008, and 2013 (45). Using this information, all trees belonging to *F. sylvatica*, *P. abies*, and *A. alba*, with a DBH between 30 cm and 100 cm, and first recorded as dead and lying in 1975, 1997, 2008, or 2013 were identified. Trees decomposing as standing before they were downed were omitted to exclude logs with unclear decay lengths. Hence, a tree species (beech, spruce, and fir), decay length (<5, 5 to 15, 16 to 38, or >38 years), and DBH was assigned to each log. Then, within each tree species and decay length class, logs were selected randomly. The final data set was composed of 118 trees: 39 beech logs (of which 9 had a decay length of <5 years, 13 of 5 to 15 years, 12 of 16 to 38 years, and 5 of >38 years), 36 fir logs (2 of <5 years, 14 of 5 to 15 years, 13 of 16 to 38 years, and 7 of >38 years), and 43 spruce logs (10 of <5 years, 12 of 5 to 15 years, 11 of 16 to 38 years, and 10 of >38 years) (see Table S2 in the supplemental material).

To obtain representative samples, four subsamples were obtained from each log in October 2013 using an electric drill with a bit diameter of 8 mm. The length of each log (or the sum of the lengths of its fragments) was measured, and subsamples were collected at one-fifth, two-fifths, three-fifths, and four-fifths of the log length. The four subsamples of each log were pooled to yield one composite sample per log. Drilling was performed vertically from the middle of the upper surface to a depth of 40 cm. The drill bit was sterilized between drillings.

 

mSystems®

**Sample processing, chemical analysis, and DNA extraction and amplification.** Details of sample processing and analysis were given in previous publications (33, 36). Briefly, the sawdust material was weighed, freeze-dried, and milled using an Ultra Centrifugal Mill ZM 200 (Retsch, Germany). Dry mass content was measured as a loss of mass during freeze-drying, and the pH was measured in distilled water (1:10). The wood C and N contents were measured in an external laboratory (Research Institute for Soil and Water Conservation, Prague, Czech Republic) as described previously (46). Klason lignin content was measured as dry weight of solids after hydrolysis with 72% (wt/wt) $H_2SO_4$ (47).

Total genomic DNA was extracted from $2 \times 200$ mg of material of each sample using the NucleoSpin Soil kit (Macherey-Nagel, Germany) according to the manufacturer's instructions. Then, PCR amplifications were performed in three PCRs per sample as described previously (33, 36) using barcoded gITS7 and ITS4 primers targeting fungal ITS2 (48) and barcoded 515F and 806R primers targeting the V4 region of the bacterial 16S rRNA gene (49). Amplicons were purified, pooled, and sequenced on the Illumina MiSeq to obtain pair-end sequences of $2 \times 250$ bp.

**Sequence data processing.** The sequencing data were processed using SEED v 2.0.3 (50) as described in references 33 and 36. For bacteria, pair-end reads were merged using fastq-join (51). For fungi, only forward read sequences beginning with the primer gITS7 were considered, since for certain highly abundant wood-decomposing fungi (e.g., *Armillaria* spp.), ITS2 is longer than 550 bases and these sequences would be missed during pair-end joining. The whole or partial ITS2 was extracted from fungal amplicons using ITS Extractor 1.0.8 (52). Sequences of inferior quality (mean Phred score of <30, all sequences with ambiguous bases) or length (<40 bases) were removed. Chimeric sequences were detected and deleted using UCHIME implementation in USEARCH 7.0.1090 (53). Sequences were clustered using UPARSE implemented in USEARCH (54) at a 97% similarity level. Consensus sequences were constructed for each cluster, and the closest hits at the species level were identified using BLASTn against UNITE (55) and GenBank for fungi and Ribosomal Database Project (56) and GenBank for bacteria. The minimum and maximum read counts were 1,598 and 21,375 for fungi, and 1,606 and 15,113 for bacteria, respectively. This resulted in 4,519 fungal and 21,260 bacterial OTUs, of which 263 and 11,601 were global singletons and were removed. Therefore, the final data consisted of 4,256 fungal and 9,659 bacterial OTUs.

**Statistical analyses.** We analyzed the data with a hierarchical modeling of species communities (HMSC) framework (57, 58), which belongs to the class of JSDM (5). In HMSC, community data are analyzed by constructing a hierarchical model in the generalized linear model (GLM) framework and using Bayesian inference.

The response data consisted of abundances (sequence counts) of bacterial and fungal OTUs in the $n = 118$ logs (i.e., sampling units). As the data were zero inflated (i.e., dominated by species' absences), we applied a hurdle model. A hurdle model consists of two parts, one modeling the presence-absence and the other modeling abundance conditional on presence. To fit the first model, we first truncated the data to presence-absence, keeping all zeros as zeros and setting the nonzeros to one, and we fitted a binomial model with probit link function to each column (i.e., OTU). Then, to fit the second model, we generated a second data set by setting all zeros to missing values (i.e., ignoring them) and keeping all nonzeros in their original values. To model these abundances conditional on presence (scaled to zero mean and unit variance), we used the lognormal model. We included both of the presence-absence and the abundance models in the same model, so that the response matrix **Y** (for notation, see reference 58) included each fungal and bacterial OTU twice. See reference 3 for more details about HMSC models, including how hurdle models can be used to model zero-inflated data.

As data on rare species are not sufficiently informative to enable fitting species-specific models (see, e.g., reference 3), we included in the analyses only those OTUs with a prevalence of >10% among the sampling units and which had at least 0.5% relative abundance in one of the sampling units. These choices resulted in 452 fungal OTUs and 570 bacterial OTUs. To test the robustness of the results against these choices, we also ran an alternative analysis, where we used 20% as the prevalence threshold and 5% as the maximal relative abundance threshold, resulting in 103 fungal and 51 bacterial OTUs. Additionally, to assess how the filtering criteria affected fungal and bacterial community structure, we carried out NMDS ordination plots separately for (i) the whole community (4,256 fungal and 9,659 bacterial OTUs); (ii) the data obtained with the filter generating a bigger data set (452 fungal and 570 bacterial OTUs); and (iii) the data obtained with filter generating the smaller data set (103 fungal and 51 bacterial OTUs). NMDS was performed using metaMDS() function from vegan (59). Because fungal communities were too complex to obtain reliable NMDS ordinations in two dimensions, we conducted the ordinations in three dimensions by setting the argument k = 3 in the metaMDS() function. The first two axes are shown in results, since interpreting three-dimensional plots is very difficult. Significant explanatory variables were added to the ordinations as arrow vectors using envfit() function from vegan (59). Significance of each explanatory variable was tested independently using 999 permutations.

As fixed explanatory variables in the matrix **X** of HMSC, we included variables related to wood chemistry and physical characteristics. The continuous variables describing wood chemistry were (i) percent water content, (ii) pH, (iii) percent C content, (iv) log-transformed percent N content, and (v) percent lignin content. Variables related to log physical characteristics were (vi) tree species (categorical with beech, spruce, and fir levels), (vii) decay time (categorical with <5, 5 to 15, 16 to 38, or >38 years levels), and (viii) DBH in centimeters (continuous). To control for variation in sequencing depth, we also included the (ix) log-transformed number of reads as a continuous variable. To identify association networks within fungi and within bacteria, as well as among these two groups, we included a community-level random effect (implemented with the help of latent variables [see reference 58]) at the sampling unit (i.e., the log) level. The community-level random effect models covariation among the species that cannot be attributed to the environmental variables, including the effects of species interactions, or to the responses of the

species to environmental covariates not included in the model (3). In the following, we refer to such unexplained covariation among the species as species associations, irrespective of what is the causal reason behind it. In an exploratory analysis prior to model fitting, we excluded C/N ratio as an explanatory variable, since it almost perfectly correlated with wood N content. We then partitioned the explained variance into fixed effects (i.e., explanatory covariates) and log-level random effects (i.e., residual association matrix). Fixed effects were further grouped as log chemical characteristics (pooling water, pH, C, N, and lignin) and log physical characteristics (pooling tree species identity, decay time, and DBH).

We built species-to-species association matrices using the correlation matrix **R** both for a model including the explanatory variables described above (called the full model) and a model which otherwise had the same structure but did not include explanatory variables (called the null model). The association matrix derived from the null model showed a combination of co-occurrence patterns created by shared habitat use as well as the patterns resulting from interactive reciprocal effects. In contrast, association matrices derived from the full model showed the co-occurrence patterns once the environmental effects were taken into account and thus were more likely to result from interactive associations.

Explanatory and predictive powers of the models were assessed by calculating AUC for presence-absence data and the standard $R^2$ for the abundance data. When computing explanatory power, models were fitted to all data, i.e., the same data were used to fit the models and make the predictions. When computing predictive power, we applied a 2-fold cross-validation approach across the sampling units. In cross-validation, sampling units are randomly divided into two folds, and to make predictions in one fold (i.e., the testing set), the models are fitted to the data from the other fold (i.e., the training set). First, we computed unconditional predictive power where community data Y is assumed to be known in only those sampling units that belong to the training set. Hence, the unconditional predictive power only uses the information on the environmental predictors and ignores the information in the species association matrix. Then, to examine the link between fungi and bacteria, we asked how much improvement the JSDM was able to make for predictions on bacterial communities when knowing the fungal composition, and, vice versa, how much improvement the JSDM was able to make for predictions on fungal communities when knowing the bacterial composition. To do so, we computed conditional predictive power, where the occurrences and abundances of bacteria were assumed to be known in the testing set when predicting fungi and where occurrences and abundances of fungi were assumed to be known when predicting bacteria. In contrast to unconditional predictive power, in conditional predictive power, the information in the species association matrix is used to make the predictions in addition to the information on environmental predictors (see reference 3 for technical details). We quantified the information value of fungi for predicting bacteria (and vice versa, the information value of bacteria for predicting fungi) as the difference between conditional and unconditional predictive power, i.e., the predictive power that did versus the power that did not utilize information about the occurrences and abundances of the non-focal species group.

We fitted the model with the R-package (60) Hmsc (57) assuming the default prior distributions. We sampled the posterior distribution with four Markov Chain Monte Carlo (MCMC) chains, each of which was run for 150,000 iterations, of which the first 50,000 were removed as burn-in. The iterations were thinned by 100 to yield 1,000 posterior samples per chain and thus 4,000 posterior samples in total. We assessed MCMC convergence by computing the effective number of samples and the potential scale reduction factor (57) for the parameters measuring species responses to environmental covariates and species-to-species associations (Fig. S5).

**Data availability.** Raw sequence data for fungi and bacteria are available in the MG RAST public database with data set numbers mgp18370 and mgp82275, respectively. Processed OTU and metadata tables as well as R scripts supporting the results have been archived in the Dryad repository (https://doi.org/10.5061/dryad.sxksn030s).

## SUPPLEMENTAL MATERIAL

Supplemental material is available online only.

**FIG S1**, TIF file, 0.1 MB.
**FIG S2**, TIF file, 0.4 MB.
**FIG S3**, TIF file, 0.8 MB.
**FIG S4**, TIF file, 0.2 MB.
**FIG S5**, TIF file, 0.1 MB.
**TABLE S1**, DOCX file, 0.1 MB.
**TABLE S2**, DOCX file, 0.1 MB.

## ACKNOWLEDGMENTS

This work was supported by the Czech Science Foundation (17-20110S). I.O. was supported by the Ministry of Education, Youth and Sports of the Czech Republic and ESIF (CZ.02.2.69/0.0/0.0/16_027/0007990).

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
