## [Reviewer comments · mSystems]

Fungal communities are important determinants of bacterial community composition in deadwood

Inaki Odriozola, Nerea Abrego, Vojtěch Tláškal, Petra Zrůstová, Daniel Morais, Tomas Vetrovsky, Otso Ovaskainen, and Petr Baldrian

Corresponding Author(s): Inaki Odriozola, Institute of Microbiology ASCR

Review Timeline:

Submission Date:	October 5, 2020
Editorial Decision:	November 30, 2020
Revision Received:	December 5, 2020
Accepted:	December 7, 2020

Editor: Ashley Shade

Reviewer(s): Disclosure of reviewer identity is with reference to reviewer comments included in decision letter(s). The following individuals involved in review of your submission have agreed to reveal their identity: Eleonora Egidi (Reviewer #1)

Transaction Report:

DOI: <https://doi.org/10.1128/mSystems.01017-20>

November 30, 2020

Dr. Inaki Odriozola
Institute of Microbiology ASCR
Prague
Czech Republic

Re: mSystems01017-20 (Fungal community composition is a better predictor of bacterial community composition than bacteria of fungal community composition in deadwood)

Dear Dr. Inaki Odriozola:

Thank you for submitting your work to mSystems. Both reviewers have agreed that the article is interesting, well-written, and a good fit for the journal, and so the work is conditionally accepted with a request for minor modifications.

In your revision, please take care to address the analysis comments of reviewer #2 - note that the reviewer does not require changes to the analysis, but rather minimally asks that the use of relativized data with the JSDM is discussed with recommendations posed for moving forward.

Below you will find the comments of the reviewers.

To submit your modified manuscript, log onto the eJP submission site at <https://msystems.msubmit.net/cgi-bin/main.plex>. If you cannot remember your password, click the "Can't remember your password?" link and follow the instructions on the screen. Go to Author Tasks and click the appropriate manuscript title to begin the resubmission process. The information that you entered when you first submitted the paper will be displayed. Please update the information as necessary. Provide (1) point-by-point responses to the issues raised by the reviewers as file type "Response to Reviewers," not in your cover letter, and (2) a PDF file that indicates the changes from the original submission (by highlighting or underlining the changes) as file type "Marked Up Manuscript - For Review Only."

Due to the SARS-CoV-2 pandemic, our typical 60 day deadline for revisions will not be applied. I hope that you will be able to submit a revised manuscript soon, but want to reassure you that the journal will be flexible in terms of timing, particularly if experimental revisions are needed. When you are ready to resubmit, please know that our staff and Editors are working remotely and handling submissions without delay. If you do not wish to modify the manuscript and prefer to submit it to another journal, please notify me of your decision immediately so that the manuscript may be formally withdrawn from consideration by mSystems.

Corresponding authors may join or renew ASM membership to obtain discounts on publication fees. Need to upgrade your membership level? Please contact Customer Service at

Service@asmusa.org.

Sincerely,

Ashley Shade

Editor, mSystems

Journals Department
Reviewer comments:

Reviewer #1 (Comments for the Author):

This study investigates the association within and between fungi and bacteria inhabiting decaying wood from a natural forest site, in an attempt to assess whether such associations result from shared habitat use or interactive effects between the two domains. To quantify the role of inter-kingdom associations, the authors partialled out the effect of contextual features (e.g., log chemistry) on bacterial and fungal co-occurrences using a joint species distribution modeling approach, considering the residual unexplained variation as the result of species associations. Then, they quantified the reciprocal role of fungal-bacterial biotic interactions in structuring the microbial communities, detecting a directional interaction between the two groups (that is, fungi explained variations in bacterial communities better than bacterial communities explained fungal occurrences).

This paper deals with a central topic in microbial ecology, i.e., the rules that govern microbial community assembly, in a relatively poorly understood system, i.e., deadwood, using a statistical approach that is well fitted to deal with non-manipulative observational data of species communities, i.e., the Hierarchical Modelling of Species Communities. This is clearly an interesting piece of work for microbial community ecologists. My main comments below refer to the need for further supporting information and some minor considerations.

Specific comments:

The title is a bit hard to understand at first glance. Consider to simplify.

Intro: To improve the flow, the paragraph on the role of fungi in deadwood decomposition (L88-98) should go before or worked into the discussion on the fungi-bacteria interaction (L74-87)

Results:

Please add details of sequencing depths and species rarefaction/accumulation curves - was the sequencing approach sufficient to capture the diversity of the two microbial communities?

L123-125: how much are the β diversity metrics correlated? It would be good to run a correlation using the Bray Curtis averages, as you did for the alpha metrics

L211: "mechanism behind" -> "underlying mechanism"

L262-263: The methods would benefit from briefly summarizing the number of replicates per decay

length/species in the main body of text. Also, I think Table S1 should be S2 and vice-versa - please check

Reviewer #2 (Comments for the Author):

Odrozola et al. presents an interesting study on co-occurring fungal and bacterial communities on deadwood, combining high-throughput sequencing data with JSDM modeling via the HMSC framework. The authors present evidence, referencing previously published work and from their modelling results, for the hypothesis that fungal-bacterial 'interactive associations' are mainly modulated through fungal communities influencing bacterial communities and not vice versa. In the present study, by combining fungal and bacterial high-throughput sequencing data, the authors tested how much the predictive power of the JSDMs for bacterial and fungal community improved when accounting for the other community. This aspect represents, in my opinion, the really interesting and novel aspect of the study. From an ecological methods perspective, there is not a lot in the literature which aims at combining data and inferring associations between organisms from two different kingdoms, such as fungi and bacteria. On top of my head, the closest link would be to studies working on bacteria and phages, and to multi-omics integration methods which tries to make inference on different 'omics' layers from the same set of samples.

I think the paper is well written. I know the authors are well-versed in the ecological modeling literature and especially in JSDMs. This is also why they are careful not to equate residual species associations directly to biotic interactions (as there has been several studies showing that co-occurrences are typically a poor proxy for biotic interactions; 1-3). At least here, the findings that the authors present seem to be in line with experimental findings. That said, given the current JSDM framework, I think the authors reach sound conclusions and do not try to over-sell their findings.

My main concern (which may or may not be addressed in this study) is that, in the best of my knowledge, JSDMs are not yet properly modelling compositional data as laid out by Aitchison (1986) (4). There are methodological advances in the field of 'microbiome science' that do so (see e.g. 5), but those seem not to have reached the JSDM field yet. The main caveat with compositional data (which I'm sure the authors are aware of) is that each sample only contains relative information. In terms of an OTU table, the total number of counts per sample is highly variable and constrained by the maximum number of DNA reads that the sequencer can provide, and this constraint induces strong dependencies among the abundances of the different taxa; e.g. an increase in the abundance of one taxon implies the decrease of the observed number of counts for some of the other taxa so that the total number of counts does not exceed the specified sequencing depth. This is why one cannot compare changes in the relative abundance even of the same taxon across samples.

While the current model (and other) partly tries to circumvent this by accounting for the sequencing depth, Aitchison (4) suggested a log-ratio transformation framework as a way explicitly deal with the aforementioned problem (4). For example, if we look at (5), which is a Multinomial Logistic-Normal Model that jointly models multiple taxa (hence, it could be interpret as a type of JSDM), the residual association matrix consists not of species covariances, but instead of log-ratios between species. I'm highlighting this here, as I think it is important for the ecological modeling community (at least the ones that have moved to work on HTS data) to follow the change that is currently happening in the microbiome field when it comes to compositional data analysis and modeling (see e.g. 4, 5; 6; 7; 8).

In the current models, are two OTU tables (where taxa are rows) of fungi and bacteria, respectively, simply stacking on top of each other? Then, yet another problem may be that a given sample (combined fungi and bacteria), in fact represent two different compositions (even if treated as one), i.e. they were not sequenced in the same run. Or is this address on line 285? If not, there are similar latent variable approaches as the one taken here (see e.g. 9) which specifically introduces a latent variable for a shared latent space between features of two different datasets (which shares the number of samples but not features).

To conclude, I do not have any objection for the publication of the current study. I do not expect the above outlined concerns to be address by these authors alone; I think it is a gradual change that has to within the whole community. I would, however, appreciate the authors authors take on the outlined problems, and if they agree that a change has to be made towards CoDa robust JSDBMs?

1. <https://onlinelibrary.wiley.com/doi/10.1111/ele.13525>
2. <https://esajournals.onlinelibrary.wiley.com/doi/10.1002/ecy.2142>
3. <https://esajournals.onlinelibrary.wiley.com/doi/full/10.1002/ecy.2133>
4. Aitchison, J. *The Statistical Analysis of Compositional Data*. (Springer Netherlands, 1986). doi:10.1007/978-94-009-4109-0.
5. <https://jsilve24.github.io/fido/articles/introduction-to-fido.html>
6. <https://www.frontiersin.org/articles/10.3389/fmicb.2017.02224/full>
7. <https://www.ncbi.nlm.nih.gov/pmc/articles/PMC6586903/>
8. <https://www.nature.com/articles/s41467-020-17041-7>
9. <https://academic.oup.com/nargab/article/2/3/lqaa050/5874365>

Reviewer comments:

Reviewer #1 (Comments for the Author):

This study investigates the association within and between fungi and bacteria inhabiting decaying wood from a natural forest site, in an attempt to assess whether such associations result from shared habitat use or interactive effects between the two domains. To quantify the role of inter-kingdom associations, the authors partialled out the effect of contextual features (e.g., log chemistry) on bacterial and fungal co-occurrences using a joint species distribution modeling approach, considering the residual unexplained variation as the result of species associations. Then, they quantified the reciprocal role of fungal-bacterial biotic interactions in structuring the microbial communities, detecting a directional interaction between the two groups (that is, fungi explained variations in bacterial communities better than bacterial communities explained fungal occurrences).

This paper deals with a central topic in microbial ecology, i.e., the rules that govern microbial community assembly, in a relatively poorly understood system, i.e., deadwood, using a statistical approach that is well fitted to deal with non-manipulative observational data of species communities, i.e., the Hierarchical Modelling of Species Communities. This is clearly an interesting piece of work for microbial community ecologists. My main comments below refer to the need for further supporting information and some minor considerations.

Specific comments:

The title is a bit hard to understand at first glance. Consider to simplify.

Answer: We have now simplified the title as follows:

Fungal communities are important determinants of bacterial community composition in deadwood

Intro: To improve the flow, the paragraph on the role of fungi in deadwood decomposition (L88-98) should go before or worked into the discussion on the fungi-bacteria interaction (L74-87)

Answer: Following reviewer's suggestion we have moved the role of fungi on deadwood decomposition to the beginning of the paragraph discussing fungal-bacterial interactions.

Results:

Please add details of sequencing depths and species rarefaction/accumulation curves - was the sequencing approach sufficient to capture the diversity of the two microbial communities?

Answer: We already added sequencing depth values in L.301-302 of methods section: "The minimum and maximum read counts were 1598 and 21375 for fungi, and, 1606 and 15113 for bacteria, respectively". Here we attach the rarefaction curves:

The figures show that most of the samples did not plateau with the sequencing depth used. However, we believe this is not crucial information to our paper. Moreover, metabarcoding is extremely vulnerable to cross-contamination of DNA between samples during sampling, DNAs extraction and library preparation. Even though one can take all measures to minimize this problem, the reality is that it is never solved and sequencing to higher depths increases the potential effect of such a carryover, when it exists. Our aim was thus not to capture the total diversity in each sample (which is unrealistic), but rather to analyse the dominating part of the community. Note that in the HMSC analyses, we only analyse species that occur at least xxx times and thus exclude the rarest species.

L123-125: how much are the β diversity metrics correlated? It would be good to run a correlation using the Bray Curtis averages, as you did for the alpha metrics

Answer: We are grateful to the reviewer for this comment. We have added the suggested analysis which shows that, as opposed to alpha diversity, beta diversity metrics are positively correlated between fungi and bacteria, using the whole communities as well as the subset communities. We added this analysis as supplementary Figure S2.

L211: "mechanism behind" -> "underlying mechanism"

Answer: Done.

L262-263: The methods would benefit from briefly summarizing the number of replicates per decay length/species in the main body of text. Also, I think Table S1 should be S2 and vice-versa - please check

Answer: We added the requested details in L.267-271. Table S2 was incorrectly labelled as S1 in L.264, now we labelled it correctly.

Reviewer #2 (Comments for the Author):

Odriozola et al. presents an interesting study on co-occurring fungal and bacterial communities on deadwood, combining high-throughput sequencing data with JSDM modeling via the HMSC framework. The authors present evidence, referencing previously published work and from their modelling results, for the hypothesis that fungal-bacterial 'interactive associations' are mainly modulated through fungal communities influencing bacterial communities and not vice versa. In the present study, by combining fungal and bacterial high-throughput sequencing data, the authors tested how much the predictive power of the JSDMs for bacterial and fungal community improved when accounting for the other community. This aspect represents, in my opinion, the really interesting and novel aspect of the study. From an ecological methods perspective, there is not a lot in the literature which aims at combining data and inferring associations between organisms from two different kingdoms, such as fungi and bacteria. On top of my head, the closest link would be to studies working on bacteria and phages, and to multi-omics integration methods which tries to make inference on different 'omics' layers from the same set of samples.

I think the paper is well written. I know the authors are well-versed in the ecological modeling literature and especially in JSDMs. This is also why they are careful not to equate residual species associations directly to biotic interactions (as there has been several studies showing that co-occurrences are typically a poor proxy for biotic interactions; 1-3). At least here, the findings that the authors present seem to be in line with experimental findings. That said, given the current

JSDM framework, I think the authors reach sound conclusions and do not try to over-sell their findings.

My main concern (which may or may not be addressed in this study) is that, in the best of my knowledge, JSDMs are not yet properly modelling compositional data as laid out by Aitchison (1986) (4). There are methodological advances in the field of 'microbiome science' that do so (see e.g. 5), but those seem not to have reached the JSDM field yet. The main caveat with compositional data (which I'm sure the authors are aware of) is that each sample only contains relative information. In terms of an OTU table, the total number of counts per sample is highly variable and constrained by the maximum number of DNA reads that the sequencer can provide, and this constraint induces strong dependencies among the abundances of the different taxa; e.g. an increase in the abundance of one taxon implies the decrease of the observed number of counts for some of the other taxa so that the total number of counts does not exceed the specified sequencing depth. This is why one cannot compare changes in the relative abundance even of the same taxon across samples.

While the current model (and other) partly tries to circumvent this by accounting for the sequencing depth, Aitchison (4) suggested a log-ratio transformation framework as a way explicitly deal with the aforementioned problem (4). For example, if we look at (5), which is a Multinomial Logistic-Normal Model that jointly models multiple taxa (hence, it could be interpreted as a type of JSDM), the residual association matrix consists not of species covariances, but instead of log-ratios between species. I'm highlighting this here, as I think it is important for the ecological modeling community (at least the ones that have moved to work on HTS data) to follow the change that is currently happening in the microbiome field when it comes to compositional data analysis and modeling (see e.g. 4, 5; 6; 7; 8).

Answer: this comment addresses important challenges in the ongoing merging of multivariate methods developed separately for microbial ecology and the community ecology of macro-organisms. As the reviewer pointed out, the JSDM approach that we applied does not model explicitly compositional data using a multinomial distribution. We consider extending the data models of JSDMs to include the multinomial distribution as one of the key future challenges, as discussed in more detail by the recent JSDM book by Ovaskainen and Abrego (Cambridge University Press; 2020). We are actually working at the moment towards the implementation of such extensions, which are in theory straightforward to define and implement, but in practice involve some non-trivial statistical and computational challenges. While the lack of the multinomial model can be viewed as a weakness of the JSDM approach, its strength (compared to most methods developed for microbial ecology) is that it accounts in a comprehensive manner for the effects of environmental covariates, species traits, phylogenies, and in particular for the structure of study designs (which can be spatial, temporal or hierarchical). Thanks to these features, it is thus less likely to result in spurious inference due to failing to account for confounding effects.

We agree that the failing to account for the multinomial nature of the data can generate in theory negative associations between species. This would be trivially the case if the data would involve two species, as then the increase in sequence count of one species would directly decrease the sequence count of the other species. However, even for the case of the two species, the presence-absence part of the model would be less problematic, as missing the occurrence of one species because the other species dominates the sequence data would be unlikely or at least require a very strong domination by the other species. Why we consider this not to be a major problem for the abundance data either is that the community consist of a very large number of species, none of which dominates the data in terms of sequence abundance. Thus the constraining effect of the total number of sequences is diluted through the whole community, and is not likely to generate strong negative associations between any pair of species. Reflecting this, the associations we report are not predominantly negative (Fig. 2), but they contain a balanced set of associations that are positive, negative or not statistically supported.

In the current models, are two OTU tables (where taxa are rows) of fungi and bacteria, respectively, simply stacking on top of each other? Then, yet another problem may be that a given sample (combined fungi and bacteria), in fact represent two different compositions (even if treated as one), i.e. they were not sequenced in the same run. Or is this address on line 285? If not, there are similar latent variable approaches as the one taken here (see e.g. 9) which specifically

introduces a latent variable for a shared latent space between features of two different datasets (which shares the number of samples but not features).

Answer: the bacterial and fungal data were indeed analyzed in a single model, as was necessary to study their associations. We accounted for the difference between these types of data by controlling for the data-type specific sequencing depth, and by having as a "species trait" the categorical variable of the data type. We further note that the latent variable model of JSMD includes multiple latent variables. Thus, if the data would not support shared latent variables between the bacteria and the fungi, it could assign e.g. one latent variable for fungi and another one for bacteria. Our results however show strong associations also between these two data types, and thus the latent variables are at least partially shared between them. Referring to the earlier comment, we note further that performing the sequencing separately for bacteria and fungi can be considered as a strength in the sense that these two datasets are independent of each other in terms of the constrained set by the total sequencing depth. Thus the negative associations estimated between bacteria and fungi cannot be an artefact due to the approach not involving the multinomial data model.

To conclude, I do not have any objection for the publication of the current study. I do not expect the above outlined concerns to be address by these authors alone; I think it is a gradual change that has to within the whole community. I would, however, appreciate the authors authors take on the outlined problems, and if they agree that a change has to be made towards CoDa robust JSMDs?

Answer: we thank very much for this. As discussed above, we consider our results to be robust even if the JSMD applied here does not include the multinomial data model, but yet agree that a key priority for the further developments of JSMDs is to implement this particular data model.

We have included a paragraph summarizing this discussion in L.231-248 of the main manuscript:

"Finally, the JSMD approach that we applied does not model explicitly compositional data using a multinomial distribution. Extending the data models of JSMDs to include the multinomial distribution is one of the key challenges in the ongoing merging of multivariate methods developed separately for microbial ecology and the community ecology of macro-organisms (3). Failing to account for the multinomial nature of the data could theoretically generate negative associations between species. This would be certainly the case if the data would involve two species, as then the increase in sequence count of one species would directly decrease the sequence count of the other species. However, even for the case of the two species, the presence-absence part of the model would be less problematic, as missing the occurrence of one species because the other species dominates the sequence data would be unlikely. Moreover, we believe that this is not a major problem for the abundance data either, since the community consists of a very large number of species, none of which dominates the data in terms of sequence abundance. Thus, the constraining effect of the total number of sequences is diluted through the whole community, and is not likely to generate strong negative associations between any pair of species. Reflecting this, the associations we report are not predominantly negative (Fig. 2), but they contain a balanced set of associations that are positive, negative or not statistically supported. Therefore, we consider our results robust even if the JSMD applied here does not include the multinomial data model."

December 7, 2020

Dr. Inaki Odriozola
Institute of Microbiology ASCR
Prague
Czech Republic

Re: mSystems01017-20R1 (Fungal communities are important determinants of bacterial community composition in deadwood)

Dear Dr. Inaki Odriozola:

Thanks for your submission to mSystems!

Your manuscript has been accepted, and I am forwarding it to the ASM Journals Department for publication. For your reference, ASM Journals' address is given below. Before it can be scheduled for publication, your manuscript will be checked by the mSystems senior production editor, Ellie Ghatineh, to make sure that all elements meet the technical requirements for publication. She will contact you if anything needs to be revised before copyediting and production can begin. Otherwise, you will be notified when your proofs are ready to be viewed.

Sincerely,

Ashley Shade
Editor, mSystems

Journals Department
Supplemental Material Fig. S2: Accept
Supplemental Material Table S1: Accept
Supplemental Material Table S2: Accept
Supplemental Material Fig. S3: Accept
Supplemental Material Fig. S5: Accept
Supplemental Material Fig. S4: Accept
Supplemental Material Fig. S1: Accept